# Adult neurogenesis through glial transdifferentiation in a CNS injury paradigm

Sergio Casas-Tinto[1,2]*, Nuria Garcia-Guillen[1], María Losada-Perez[1,3]*

[1]Instituto Cajal (CSIC), Madrid, Spain; [2]Instituto de Investigación de Enfermedades Raras (IIER-ISCIII), Majadahonda, Spain; [3]Universidad Complutense de Madrid (UCM), Madrid, Spain

## eLife Assessment

In this work, the authors use a *Drosophila melanogaster* adult ventral nerve cord injury model extending and confirming previous observations. This **important** study reveals key aspects of adult neural plasticity. Taking advantage of several genetic reporter and fate tracing tools, the authors provide **solid** evidence for different forms of glial plasticity, that are increased upon injury. The significance of the generated cell types under homeostatic conditions and in response to injury remains to be further explored and open up new avenues of research.

**\*For correspondence:**
sergio.casas@isciii.es (SC-T);
marilosa@ucm.es (ML-P)

**Competing interest:** The authors declare that no competing interests exist.

## Abstract

As the global population ages, the prevalence of neurodegenerative disorders is fast increasing. This neurodegeneration as well as other central nervous system (CNS) injuries cause permanent disabilities. Thus, generation of new neurons is the rosetta stone in contemporary neuroscience. Glial cells support CNS homeostasis through evolutionary conserved mechanisms. Upon damage, glial cells activate an immune and inflammatory response to clear the injury site from debris and proliferate to restore cell number. This glial regenerative response (GRR) is mediated by the neuropil-associated glia (NG) in *Drosophila*, equivalent to vertebrate astrocytes, oligodendrocytes (OL), and oligodendrocyte progenitor cells (OPCs). Here, we examine the contribution of NG lineages and the GRR in response to injury. The results indicate that NG exchanges identities between ensheathing glia (EG) and astrocyte-like glia (ALG). Additionally, we found that NG cells undergo transdifferentiation to yield neurons. Moreover, this transdifferentiation increases in injury conditions. Thus, these data demonstrate that glial cells are able to generate new neurons through direct transdifferentiation. The present work makes a fundamental contribution to the CNS regeneration field and describes a new physiological mechanism to generate new neurons.

## Introduction

Neuronal loss caused by injury or neurodegenerative diseases is irreversible in the central nervous system (CNS) of vertebrates. Spinal fractures represent 5–6% of all fractures, and spinal cord damage complicates 16–40% of spinal fractures causing significant impairment to patients (***Ding et al., 2022***). Moreover, according to the United Nations, the prevalence of neurodegenerative diseases will almost double in the next 20 years. Current therapeutic strategies aim to replace damaged or lost cells through dedifferentiation and transplantation or cell reprogramming of glial cells. Thus, understanding glial cell contribution after CNS damage is relevant to attain its functional recovery (***Li et al., 2017***).

Glial cells activate inflammatory and proliferative responses upon injury in a process known as glial regenerative response (GRR) (***Kato et al., 2018***; ***Losada-Perez, 2018***). This response is described

in multiple species of vertebrates and invertebrates, and the molecular mechanisms underlying this process are conserved from *Drosophila* to mice or humans (*Chandra et al., 2023*; *Kato et al., 2018*; *Kato et al., 2015*; *Kato et al., 2011*; *Li et al., 2020*; *Losada-Perez, 2018*; *Losada-Pérez et al., 2021*; *Purice et al., 2017*; *Smith et al., 1987*). Glial proliferation is a well-known cellular response upon CNS damage (*Chandra et al., 2023*; *Kato et al., 2015*; *Kato et al., 2011*; *Li et al., 2020*; *Li and Hidalgo, 2020*; *Purice et al., 2017*), however, it is not clear how this response contributes to CNS regeneration. In mammals, oligodendrocyte progenitor cells (OPCs or NG2-glia) react, giving rise to oligodendrocytes (OL) while in *Drosophila,* neuropil-associated glia play that role (*Losada-Pérez et al., 2021*). Upon injury, mammalian OPCs can produce astrocytes and OL contributing to CNS repair (*Kato et al., 2018*; *Niu et al., 2021*). GRR of neuropil glia (NPG) in *Drosophila* larva is modulated by a genetic network that controls the switch between quiescence, proliferation, and differentiation. However, whether these mechanisms operate in the mature differentiated adult brain remains unknown. In *Drosophila*, NPG somata reside in the cortex/neuropil interface (*Figure 1A*). NPG includes two groups: astrocyte-like glia (ALG) and ensheathing glia (EG) (*Kremer et al., 2017*; *Losada-Perez, 2018*; *Shweta Sharma et al., 2024*). EG enwraps the neuropil and bundles of axons within the neuropil that define different functional regions (*Losada-Perez, 2018*). By contrast, ALG cells project to the neuropil forming a dense meshwork with the synapses (*Losada-Perez, 2018*), which suggests that ALG cells may play a significant role in maintaining the architecture and function of the neuropil. Larval ALG upregulates *kon-tiki* expression upon injury, a gene which is required for glial division (*Losada-Perez et al., 2016*). Besides, larval EG during metamorphosis can dedifferentiate, proliferate, and redifferentiate into adult ALG and EG (*Kato et al., 2020*). Thus, NPG appear as good candidates to replace lost cells with functional ones.

We have studied the physiological mechanisms that occur after CNS injury, focusing on the fate of glial cells upon damage. We provide the first evidence of in vivo physiological reprogramming of glial cells into other glial identities and into new neurons in the adult brain.

## Results
### ALG proliferation in adult is increased upon crush injury

Glial cells play a crucial role in the regenerative processes of the nervous system by providing structural support, modulating inflammation, and facilitating tissue repair. They secrete molecular signals that influence neuronal survival, axonal regrowth, and synaptic remodelling (*Gallo and Deneen, 2014*; *Losada-Pérez et al., 2021*). These signals include cytokines, growth factors, and extracellular matrix components, which coordinate the activation of repair pathways and the recruitment of immune cells to the injury site.

Understanding the precise molecular mechanisms that regulate glial cell behaviour is critical for developing therapeutic strategies aimed at enhancing functional recovery after nervous system injuries. Previously we described that glial cells proliferate in basal conditions and in response to crush injury in the adult fly (*Losada-Pérez et al., 2021*), although we did not identify the glial subtype responsible for this response. Thus, we questioned if those dividing cells were of the NPG type, i.e., ALG, as previous works suggest in larvae (*Kato et al., 2020*; *Losada-Perez et al., 2016*), or EG.

ALG identity is defined by the co-expression of *repo* and *prospero* transcription factors (*Omoto et al., 2015*). To determine if ALG cells divide upon injury, we caused a crush in the metathoracic neuromere of the adult ventral nerve cord (VNC, *Figure 1A*; *Losada-Pérez et al., 2021*) and performed EdU experiments combined with staining with anti-Repo (pan-glia marker) and anti-Prospero antibodies. The results show that ALG cells (Repo positive, Prospero positive) divide (EdU positive) after injury (*Figure 1B*). Besides, we also observed EdU+ Repo+ cells that were Pros negative. Therefore, this result indicates that glial cells, different from ALG, also undergo division.

There is no specific nuclear marker for EG cells, consequently, to determine if the other dividing cells were EG, we expressed the fluorescent ubiquitination-based cell cycle indicator FLY-FUCCI (*Zielke et al., 2014*) under control of a specific driver for EG (R56F03-Gal4). This genetic tool allows the visualisation and monitorisation of cell cycle-regulated proteins tagged with fluorescent markers, allowing to determine the cell cycle phase of individual cells based on their fluorescence (see Materials and methods). In parallel, to monitor cell cycle progression in ALG cells, we expressed FLY-FUCCI

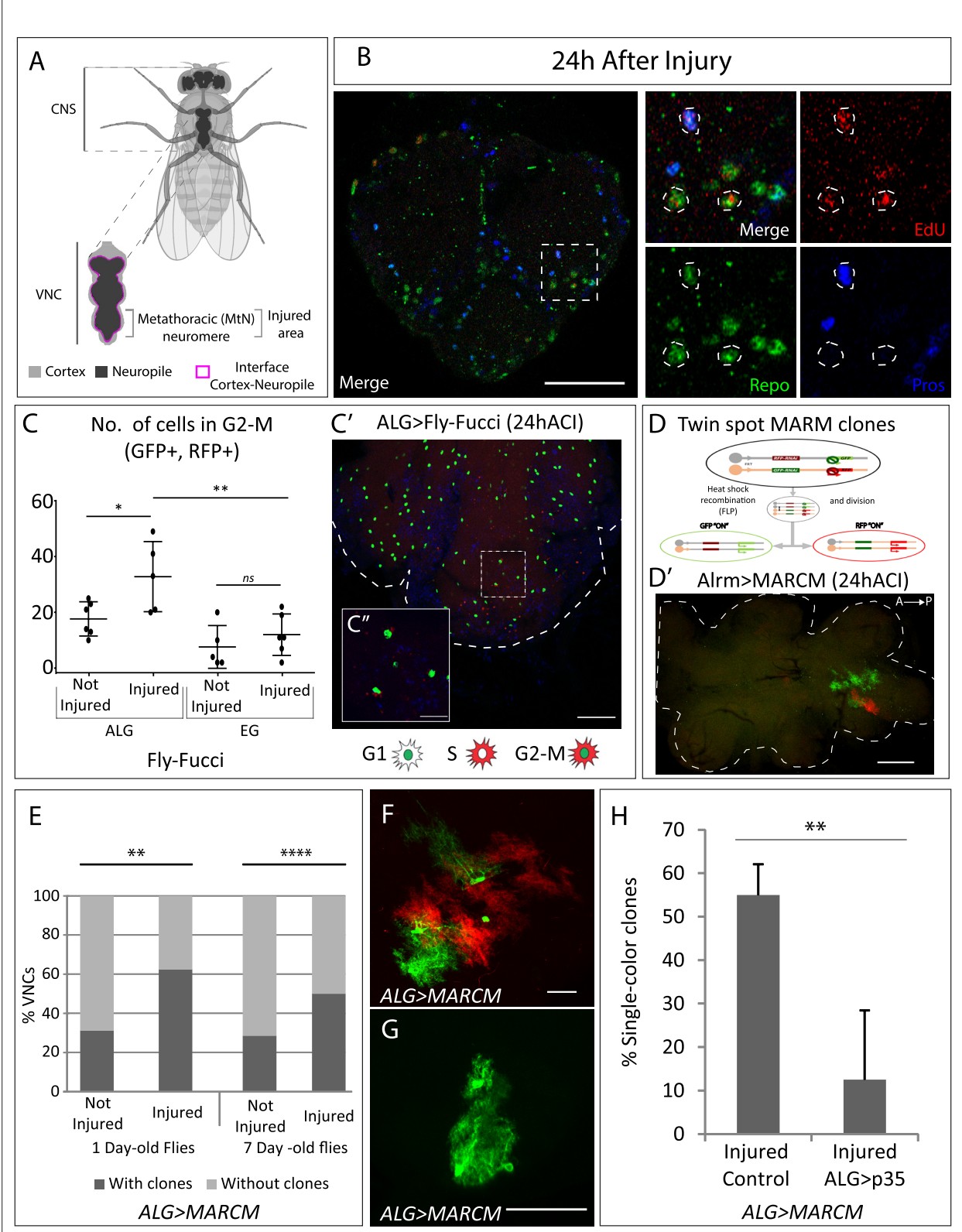

**Figure 1.** Neuropil glia proliferation in normal conditions and upon crush injury in adults. (**A**) Schematic representation of *Drosophila* adult central nervous system (CNS) and ventral nerve cord (VNC), indicating the MtN, where the injury is performed. (**B**) Representative confocal image showing dividing cells (EdU+) with astrocyte-like glia (ALG) identity (Repo+ and Pros+) and glial identity different from ALG (Repo+, Pros-). (**C**) Number of ALG or ensheathing glia (EG) cells in G2-M per MtN, in normal conditions and 24 hr after injury. Paired t-test, *p<0.05, **p<0.01. (**C′–C′′**) Representative

*Figure 1 continued on next page*

*Figure 1 continued*

images of ALG in G2-M showing their position within the injured area (**C'**) and a magnification showing the detail of GFP+RFP+ cells (**C"**). (**D**) Diagram of how the twin-spot mosaic analysis with a repressible cell marker (MARCM) tool works. (**D'**) Representative confocal image of ALG clones generated with twin-spot MARCM. (**E**) Number of VNCs with/without clones in controls and injured VNCs in young (1-day-old) and mature (7-day-old) animals. Chi-square binomial test, **p<0.01, ****p<0.0001. (**F–G**) Representative confocal images of a two-colour ALG clone (**F**) and a single-colour ALG clone (**G**). (**H**) Percentage of ALG single-colour clones in injured controls and injured VNCs where apoptosis was inhibited in ALG. Unpaired t-test, **p<0.01. Genotypes: wild-type (**B**); *AlrmGal4>UAS-Fly-FUCCI* (**C-C'**); *R56F03Gal4>UAS-Fly-FUCCI* (**C**); *HsFLP; UAS-GFP, UAS-RFP$_{RNAi}$ /UAS-RFP, UAS-GFP$_{RNAi}$; tubGal80ts:alrmGal4;* (**D'–H**) *HsFLP; UAS-GFP, UAS-RFP$_{RNAi}$ /UAS-RFP, UAS-GFP$_{RNAi}$; tubGal80ts:alrmGal4/UAS-p35* (**H**). Scale bars: 50 μm (**B, C, E**); 15 μm (**C", F, G**).

The online version of this article includes the following source data for figure 1:

**Source data 1.** Nuclear sizes and position of transdifferentiated glial cells.

under control of the ALG-specific driver alrm-Gal4 (*Altenhein et al., 2006*; *Beckervordersandforth et al., 2008*).

We quantified the number of ALG or EG cells in G2-M (i.e. GFP and RFP positive) in control and injured animals at 24 hr after injury. The data show that EG cells have a basal division rate which does not change in response to injury. In contrast, ALG cells have a higher division rate in basal conditions compared to EG, and this rate increases in response to injury (*Figure 1C–C'*). This result indicates that adult ALG cells activate a proliferative response after injury, akin to the proliferative response in larvae (*Losada-Perez et al., 2016*). On the other hand, the basal proliferation of EG explains the presence of Repo+ EdU+ Pros- cells.

To further confirm these results, we generated twin-spot mosaic analysis with a repressible cell marker (MARCM) clones (*Yu et al., 2009*) using specific drivers for EG (R65F03-Gal4) or ALG (alrm-Gal4). We combined the twin-spot MARCM technique with tubGal80$^{TS}$ under temperature control (see Materials and methods). This technique allowed the study of ALG or EG division selectively in adulthood (*Figure 1D*).

The results of ALG MARCM clones indicate that ALG divides in normal conditions in the adult CNS. Since ALG cells are very heterogeneous in size and shape, we were not able to determine the number of cells that composed each clone (*Figure 1D'*). However, we quantified the number of individuals with clones. The results indicate that the percentage of samples with clones is significantly higher in injured animals compared to controls (*Figure 1E*). To confirm that this division is not the result of residual development, we repeated the experiment with *mature* (7-day-old) flies. In this case, the percentage of not-injured CNS with clones is lower compared with young flies (*Figure 1E*), indicating we were observing a developmental effect. However, in *mature* flies, the percentage of samples with clones is significantly higher in injured animals compared to not-injured controls (*Figure 1E*), indicating that ALG divide upon damage after the developmental program is finished.

MARCM clones induction with the specific driver for EG did not generate any clone, neither in normal conditions nor after crush injury (ACI). MARCM technique generates clones in a stochastic manner, therefore, the lack of MARCM clones does not indicate that EG never divides, instead, it indicates that this event is less frequent than ALG division. These results are consistent with the FLY-FUCCI results.

Altogether, the results indicate that there is a basal division rate of neuropil glia where EG divides less often than ALG. Also, ALG shows a proliferative response to crush injury. These findings go in line with previous reports indicating the proliferative response of glial cells (*Kato et al., 2018*; *Losada-Perez, 2018*). Moreover, the results now describe proliferation of adult fully differentiated glial cells and define a differential response of EG and ALG in basal conditions, and upon injury. OPCs carry the proliferative response upon injury in mammals (*Kato et al., 2018*; *Losada-Perez, 2018*). Therefore, these results further support the homology between ALG and vertebrate OPC.

## Programmed cell death regulates ALG number after injury

The twin-spot MARCM technique implies that the cell division includes homologous recombination. The first division would give rise to two daughter cells, one GFP positive and other RFP positive (*Figure 1D*). These differentially marked daughter cells could further divide or not, yielding clones of different sizes (*Figure 1D' and F*). However, during ALG MARCM clone quantification, we noticed that >50% of clones were single-colour clones (*Figure 1G and H*). This means that after the first

division, one of the daughter cells either dies or changes its identity (as we use the alrm-Gal4 ALG-specific driver to express GFP or RFP). To determine if single-colour clones could be explained by programmed cell death (PCD), we inhibited it by co-expressing UAS-p35 (*Hay et al., 1994*), while inducing twin-spot MARCM clones in injured animals. The results showed that p35 significantly reduced the number of single-colour clones (*Figure 1H*). These results indicate that a portion of dividing ALG cells undergo PCD, reducing the final number of newly generated ALG after injury. This cellular behaviour is commonly observed during development, where cells are produced in excess and are eliminated by PCD later (*Fuchs and Steller, 2011*; *Gabilondo et al., 2018*; *Miguel-Aliaga and Thor, 2009*). Now we describe how differentiated glial cells reproduce this proliferative response and the elimination of cells by PCD.

## NP glia exchange identities between EG and ALG

Apoptosis inhibition significantly reduces ALG single-colour clones, but we could still observe them. This could be explained not only by the different effectiveness of the UAS- p35 to suppress apoptosis, but also by the existence of dedifferentiation or transdifferentiation processes. To test this hypothesis, we combined the G-TRACE tool, which allows the visualisation of historical and current Gal4 activation, with TubGal80ts to restrict the G-TRACE historical visualisation to adulthood.

We drove the expression of G-TRACE reporter with the specific drivers for ALG or EG and analysed the historical and current activation of ALG or EG drivers. The results show that in both ALG and EG experiments, there were cells with historical driver activation but not current activation (GFP+RFP-) (*Figure 2A–A'*). Also, those cells have distinctive nuclear sizes and positions (*Figure 2—figure supplement 1*). We named the first type neuropil-like cells (NP-like, arrowheads, *Figure 2A*), as they have similar nuclear shape and size to ALG and EG and they were found in the cortex-neuropil interface, where neuropil glia nuclei are usually positioned (*Figure 1A*). The second type of cells were found in the ventral cortex (VC) and presented larger nuclei compared to NP-like cells (*Figure 2—figure supplement 1A and B*), thus, given their position we named those cells as VC cells (arrowheads, *Figure 2A'*).

Next, we asked if those cells that changed their fate still had a glial identity. To answer this question, we stained NP-like cells with the pan-glial marker Repo. The results indicate that NP-like cells are glia as they were Repo+ (*Figure 2B–B'*). To confirm this, we expressed Gal4 technique for real-time and clonal expression (G-TRACE) under the control of *Repo Gal4* pan-glial driver and analysed the presence or absence of transformed cells. We identified GFP+ RFP- cells localised in the VC with identical nuclear shape as the VC cells identified in previous experiments (*Figure 2C*), however, we could not find any NP-like cells. Thus, it seems that NP-like cells have glial identity while VC cells have not.

Next, we addressed if the NP-like glia fate change was triggered by injury. To that end we quantified the number of NP-like cells transformed from ALG or from EG, in not-injured and injured animals 6 hr and 24 hr ACI. The results show that the number of NP-like cells does not increase in response to injury no matter their initial identity, ALG or EG (*Figure 2D*). However, we found significant differences in the number of transformed cells that originated from ALG or EG. We observed that ALG to NP-like transformation is less frequent than EG to NP-like transformation (<0.5% in ALG compared with >10% in EG) (*Figure 2E*).

The transformation of EG into ALG has been shown during metamorphosis (*Kato et al., 2020*), indicating the versatility of those cells. Thus, we use anti-prospero to determine if the NP-like cells originated from EG have ALG identity in this context. The Confocal images show that the NP-like cells originated from EG where pros+ (*Figure 3A*), suggesting that EG transdifferentiate into ALG. Next, to determine if Prospero is required for EG to ALG transformation, we quantified and compared the number of NP-like cells that arose from EG origin in controls and in individuals where *pros* was downregulated specifically in EG cells. The results showed a significant decrease in the number of NP-like cells upon pros knockdown in EG cells (*Figure 3B*). These results indicate that *pros* downregulation in EG reduces inter-glia conversion. Thus, we can conclude that Prospero is required for the EG to ALG transformation in adult CNS.

This is the first report of EG to ALG transformation in vivo in adult flies. The functional implications of this process include CNS regeneration, cellular homeostasis, and behavioural consequences including locomotion recovery, but probably also learning and memory. The concept of 'baseline' plasticity in adult tissues suggests that even in the absence of injury or external stimuli, certain cell

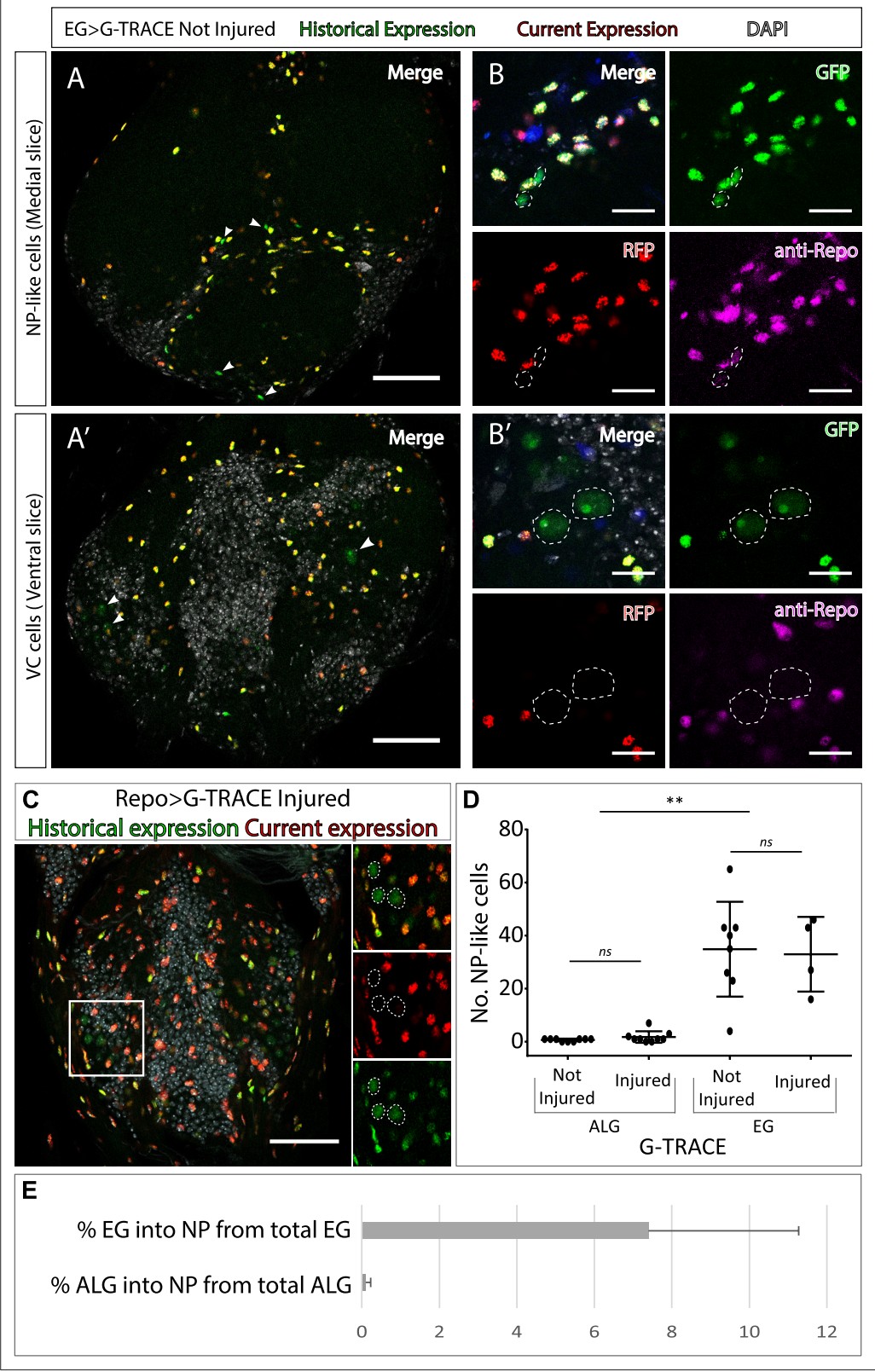

**Figure 2.** Neuropil glia transdifferentiate into two different cell types. (**A–B**) Representative confocal slices showing transdifferentiated ensheathing glia (EG) cells with the Gal4 technique for real-time and clonal expression (G-TRACE) tool. (**A**) Medial slice showing neuropil (NP)-like cells position within the MtN. (**B**) Zoom slice (merge and separated channels) showing NP-like cells (GPF+, RFP-) have glial identity (repo+). (**A′**) Ventral slice showing ventral

*Figure 2 continued on next page*

*Figure 2 continued*

cortex (VC) cells position within the MtN. (**B′**) Zoom slice (merge and separated channels) showing VC cells (GPF+, RFP-) are not glia (repo-). (**C**) Representative ventral slice showing transdifferentiated CV cells visualised with G-TRACE toll driven by the pan-glial factor Repo-Gal4. (**D**) Number of NP-like cells originated from ALG or EG in not-injured and injured MtM. Mann-Whitney test *ns* p>0.05, **p<0.01. (**E**) Percentage of NP-like cells compared with the total number of NP glia that originate them. Scale bars: 50 μm (**A**, **A′**, **C**); 15 μm (**B**, **B′**). Genotypes: *tubGal80ts, R56F03Gal4>UAS G-TRACE* (**A, B, D, E**); *tubGal80ts, repoGal4>UAS G-TRACE* (**C**); *tubGal80ts, AlrmGal4>UAS G-TRACE* (**D, E**).

The online version of this article includes the following source data and figure supplement(s) for figure 2:

**Source data 1.** Quantifications and statistic analysis for *Figure 2*.

**Figure supplement 1.** Nuclei sizes of transdifferentiated neuropil glia (NPG) and localisation of ventral cortex (VC) cells.

---

types may retain the ability to undergo phenotypic changes. In the context of glial cells, this plasticity could reflect an intrinsic readiness to adapt to the dynamic needs of the surrounding tissue, such as maintaining homeostasis, or supporting regeneration. This property of glial cells might impact on compensatory mechanisms for neurodegenerative processes, tissue engineering, and regenerative therapies that exploit endogenous repair mechanisms and cellular adaptability in the nervous system to functional needs.

## NP glia transdifferentiate into neurons upon injury

Repo-G-TRACE experiments demonstrated that VC cells are not glia (*Figure 2B and C*). To test if these new cells have a neuronal identity, we stained the VNCs with the pan-neuronal marker Elav. The analysis showed that these cells were Elav positive (*Figure 4A–A′*). This result indicates that glial cells can transdifferentiate into neurons under normal conditions. Next, we quantified the number of VC cells following injury. The results show that the number of VC cells is significantly higher in injured animals compared to not-injured controls (*Figure 4B*). Next, we analysed which NPG subpopulation

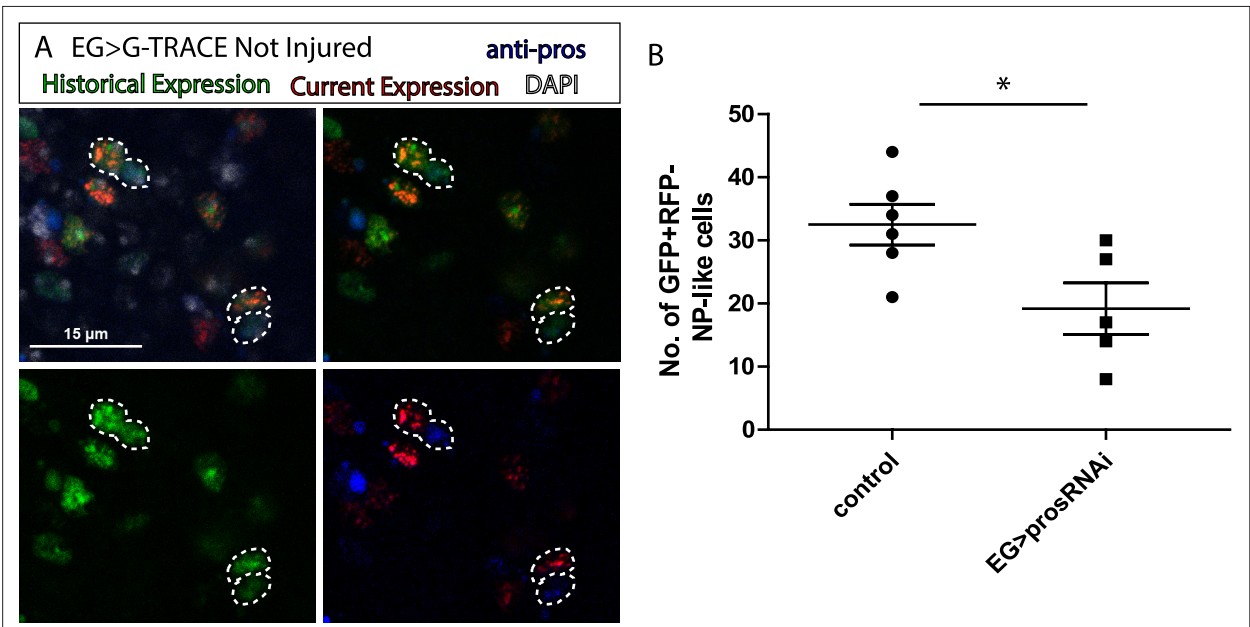

**Figure 3.** Ensheathing glia (EG) transdifferentiation into astrocyte-like glia (ALG) requires prospero. (**A**) Representative image of neuropil (NP)-like cells originated from EG stained with ALG marker Prospero. Top left: Merge image with DAPI. Top right: Merge image without DAPI. Bottom left: GFP signal. Bottom right: RFP signal and pros staining. (**B**) Number of NP-like cells originated from EG in controls animals where pros was inhibited in EG. Unpaired t-test, *p<0.05. Scale bar: 15 μm (**A**). Genotypes: *tubGal80ts, R56F03Gal4>UAS G-TRACE* (**A**, **B**).

The online version of this article includes the following source data for figure 3:

**Source data 1.** Quantifications and statistic analysis for *Figure 3*.

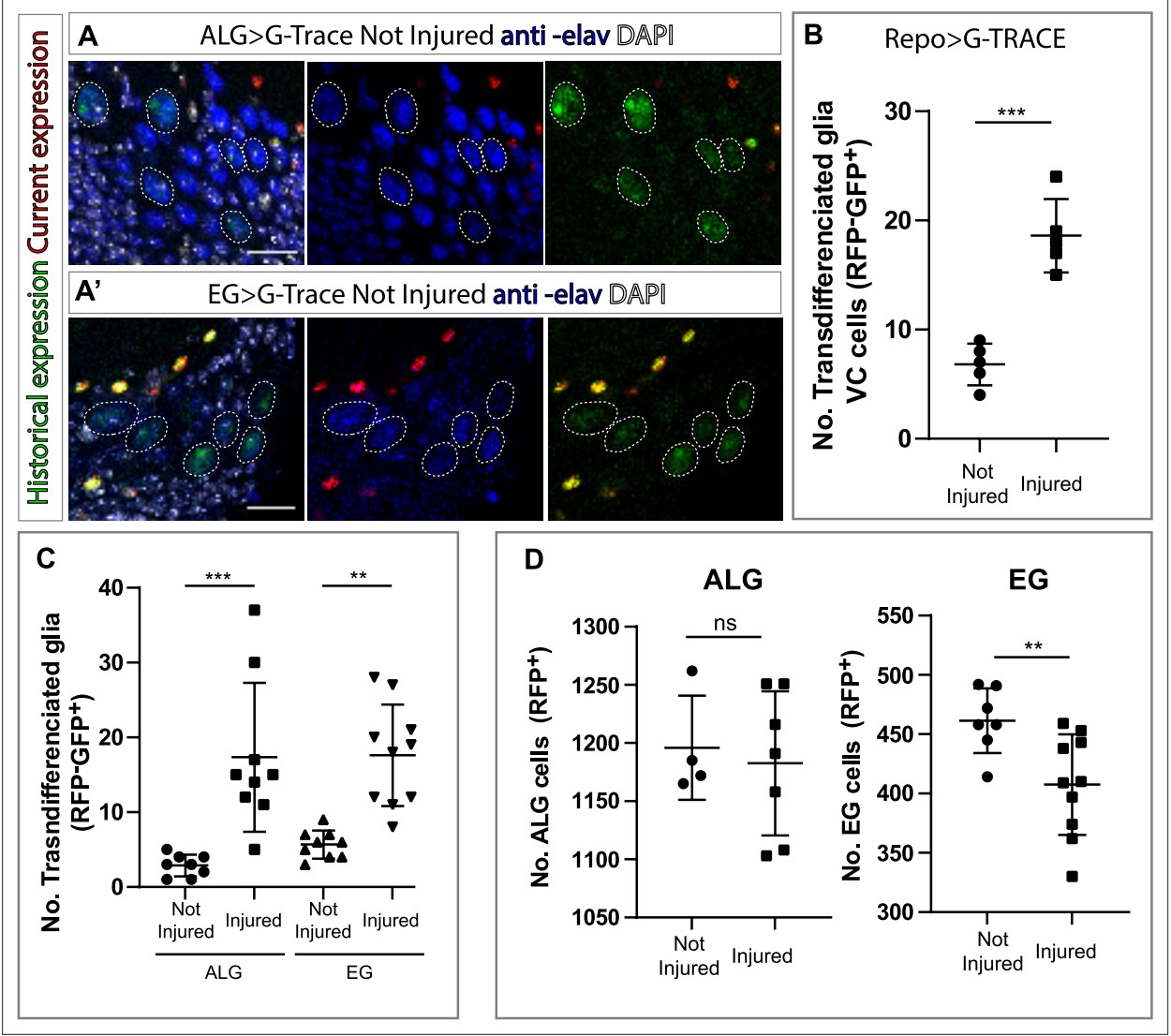

**Figure 4.** Ensheathing glia (EG) and astrocyte-like glia (ALG) transdifferentiate into neurons upon injury through two different mechanisms. (**A–A'**) Transdifferentiated ventral cortex (VC) cells (GFP+, RFP-) raised from ALG (**A**) and EG (**A'**) are Elav+. From left to right: Merge with DAPI, merge without DAPI and green and red channels merged. (**B**) Number of transdifferentiated glia (VC cells, GFP+, RFP-) in controls and injured ventral nerve cords (VNCs). (**C**) Number of transdifferentiated VC cells (GFP+, RFP-) originates from ALG or EG in controls and injured mature animals. (**D**) Number of ALG or EG that remain ALG or EG in controls and injured VNCs. Statistics (**B, C, D**): Paired t-test, ***p<0.001, **p<0.01, *ns* p>0.05. Scale bar: 15 µm (**A**). Genotypes: *tubGal80ts, R56F03Gal4>UAS G-TRACE* and *tubGal80ts, AlrmGal4>UAS G-TRACE*.

The online version of this article includes the following source data and figure supplement(s) for figure 4:

**Source data 1.** Quantifications and statistic analysis for *Figure 4*.

**Figure supplement 1.** New neurons do not co-localise with markers for mature neurons.

(i.e. ALG or EG) were responsible for this increase in VC neurons. To eliminate any developmental effect, we analysed in mature flies (see Materials and methods) the possible transdifferentiation from ALG or EG into VC in control and injured animals. The result showed that the new neurons originate from both transdifferentiated ALG and EG (*Figure 4C*).

So far, we have seen that ALG cells divide upon injury while EG do not. On the other hand, VC neurons increase from ALG and EG types in response to injury. This could indicate that ALG divides prior to transdifferentiation while EG transdifferentiate directly. To test this hypothesis, we quantified the number of ALG and EG cells in both injured animals and not-injured controls. We found a significant increase in ALG number in injured animals compared to controls in line with our previous results.

Moreover, we found a significant decrease in EG in injured animals compared to not-injured controls, supporting the direct transformation in response to injury (*Figure 4D*).

Finally, to determine the identity of transdifferentiated neurons, we stained VC cells with different neuronal markers such GABA, ChaT, or GluRIIA (*Boppana et al., 2017*; *Chen et al., 2016*; *Katheder et al., 2023*; *Kolodziejczyk et al., 2008*). The images showed that VC cells are negative for all markers (*Figure 4—figure supplement 1*), preventing definitive identification of their neuronal subtype. The absence of neuronal marker expression in VC cells suggests an immature cellular state. It is plausible that, given sufficient time and appropriate environmental cues, these cells might undergo further differentiation into mature neurons. Such maturation could enable their functional integration into existing neuronal networks, potentially contributing to nervous system regeneration. However, this hypothesis remains speculative and requires further investigation. Future studies should explore the conditions that promote VC cell maturation, including the influence of extracellular signals, transcriptional regulation, and synaptic connectivity. Understanding these processes could advance the development of regenerative therapies targeting neurodegenerative diseases or injuries.

Taken together, our results indicate that ALG and EG generate new neurons in response to injury through two different mechanisms. ALG divide and transdifferentiate while EG transdifferentiate directly. Thus, glial cells reveal as key contributors to newly generated glia and neurons during regeneration.

Many authors are exploring the potential of astrocytes to generate new neurons, however, they are focused on artificial reprogramming or transdifferentiation (*Talifu et al., 2023*). This is the first description of this phenomenon observed in vivo in wild-type animals. Therefore, these findings open a new approach to generate new neurons in a damaged CNS. Understanding the molecular mechanisms that govern this physiological glial behaviour would have an immeasurable value.

## Materials and methods

### Fly stocks and genetics

Experiments were performed in adult *Drosophila melanogaster* flies raised at 25°C or combining 17°C with 29°C for temporal control of UAS/Gal-4 using the thermosensitive repression system Gal80 TS. Fly stocks used were: *Alrm-Gal4* (BL 67032), *R56F03-Gal4* (BL 39157), *UAS-GFP.E2f; UAS-mRFP.CycB* (Fly- FUCCI, BL 55100), *repo-Gal4* (BL-7415), *tub Gal80 ts* (BL-7019), *UAS-p35* (BL5073), UAS-G-TRACE (BL 28281), Twin-Spot *MARCM stocks* (BL56184 and BL56185), *UAS-prosRNAi* (BL-26745) from Bloomington Drosophila Stock Center and *yw hs-FLP* from A Ferrús. To control the activation of different tools, we combined the UAS/Gal4 system with the *Tubulin-Gal80$^{TS}$* (temperature-sensitive) tool. We maintained the desired genotypes at non-permissive temperature (i.e. 17°C, where the Gal80$^{TS}$ repressor protein is active), to block UAS constructs expression.

### Crush injury

Crush injury was performed following the protocol described in *Losada-Pérez et al., 2021*. When the experiment did not require Gal4/UAS activation exclusively in adult (i.e. UAS Fly-Fucci and Twin-Spot MARCM), we breed flies at 25°C; we selected 1-day-old adult flies emerging between 12 and 24 hr before injury; we selected mature flies that emerged 5–7 days before injury. In the experiments where we needed to temperature control Gal4 activation (i.e. UAS-G-TRACE), we breed flies at 17°C, then we selected that emerged 24–48 hr before injury (for 1-day-old flies) or that emerged 10–12 days (for mature flies) and then collocated at permissive temperature (i.e. 29°C, where Gal80 TS is inactive and the Gal4/UAS system is active).

Injured flies dragging the third pair of legs that did not show external signs of cuticle damage were selected 24 hr later (24 hr ACI).

### FLY-FUCCI system

FLY-FUCCI (*Zielke et al., 2014*) allows the visualisation of cell cycle activity, GFP, or RFP-tagged forms of E2F1 and Cyclin B proteins respectively are expressed using the UAS/Gal4 system. Thus, by the presence/absence of GFP and RFP, it is possible to determine the cell cycle stage. We identify GFP positive, RFP positive cells as cells in the G2-M cell cycle phase.

## Twin-spot MARCM

Using twin-spot MARCM technique (*Yu et al., 2009*), it is possible to detect cells dividing in a specific moment. When a cell divides, each of the two resulting cells is marked with GFP or RFP, always considering that the promoter directing this tool gets activated. This is possible because the genetic constructs used within this tool includes a flippase under the control of a heat shock promoter (HsFlp). This enzyme induces mitotic recombination of sequences situated between flippase recognition target (FRT) sites. To induce flippase activation, flies were exposed to heat shocks by introducing the vials in 37°C water baths for 60 min.

### Twin-spot MARCM clone induction

We performed two 60-min heat shocks in a water bath at 37°C to induce twin-spot MARCM clones. Vials with 1-day-old flies received the first heat shock right before crush injury. Then, flies were placed at 25°C s for 6 hr. After that, flies received the second heat shock and placed again at 25°C. Afterwards, flies were dissected 24 hr after injury.

## G-TRACE tool

G-TRACE tool (*Evans et al., 2009*) allows the visualisation of historical and current expression of *Gal4* gene. It is based on the combination of three genetic constructs: *UAS-RFP, UAS-FLP,* and *Ubi$^{63}$-FRT-Stop-FRT-GFP.*

Whenever *Gal4* expression is activated, it induces the expression of *UAS-RFP* and *UAS-FLP*. Flippase enzyme recognises *FRT* sites and removes stop sequences, allowing *GFP* constitutive expression (GFP gene is under the control of a constitutive promoter *Ubiquitine-p63E*). Cells expressing *Gal4*, and the posterior lineage, will be marked with GFP. This tool allows tracking the current expression of a *Gal4* enhancer sequence (*UAS-RFP* in red) and its historical expression (*Ubi-GFP* in green).

### G-TRACE activation

Animals that develop at 17°C were selected 24–48 hr after emerging and were collocated at permissive temperature (29°C) for 6 hr. Then, the crush injury was performed. Flies were selected and dissected 24 hr ACI.

## EdU incorporation

We used EdU staining to detect cell proliferation. Flies were injured and left overnight in Eppendorf tubes containing EdU solution (100 μM EdU in 5% sugar, 1% dye). 24 hr after injury flies were dissected in cold 4% FA and fixed 15 min in 4% FA while rocking and washed twice with 3% bovine serum albumin (BSA) in PBS. Fixed VNCs were permeabilised with PBSTx0.5 (1×PBS, 0.5% Triton X-100) for 20 min and washed twice again with 3% BSA in PBS. EdU detection was performed by using the Click-iT reaction cocktail (Click-iT EdU, Invitrogen). Afterwards, the CNS were immunostained following the protocol described below.

## Immunohistochemistry and EdU detection

We performed the immunohistochemistry following the protocol for adult *Drosophila* immunolabelling according to *Purice et al., 2017*.

We used the following antibodies: mouse anti-Repo (DSHB, 8D12 ID: AB_528448, 1:200), mouse anti-Elav (DSHB, 9F8A9 ID: AB_528217), mouse anti-prospero (DSHB, MR1A, ID: AB_528440), mouse anti-ChAT (DSHB, ID:AB_528122), mouse anti-GluRIIA (DSHB, ID: 528269), guinea pig anti-Repo (gift from B Altenhein), anti-Ph3 (Millipore, 3H10), anti-GABA (Sigma-Aldrich, A2052; *Jackson et al., 1990*).

## Quantification, statistical analysis, and imaging

Images were acquired by confocal microscopy (LEICA TCS SP5) and processed using Fiji (ImageJ 1.50e) software. Images were assembled using Adobe Photoshop CS4 and Adobe Illustrator CS4. Cells were counted using the cell counter plugin from Fiji or Imaris (Imaris 6.3.1 software - Bitplane). Areas were measured with measurements plugin from Fiji. Data were analysed and plotted using GraphPad Prism v7.0.0 or v9.0.0. For qualitative data, analysed in *Figure 1D–D'*, we performed a binomial test (**p<0.01). For quantitative data, we used a D'Agostino-Pearson normality test and

analysed data with normal distributions by using two-tailed t-test with Welch's unequal variances t-test. Error bars represent standard deviation (±s.d.).

## Acknowledgements

We thank Prof. Alberto Ferrús, Dr Francisco Martín, Prof. Agustín Zapata, and the anonymous reviewers for critiques of the manuscript and helpful discussions and Carlos Rodriguez and Esther Seco for flystocks maintenance. We are grateful to the Bloomington *Drosophila* Stock Centre and the Developmental Studies Hydridoma Bank and Benjamin Altenhein for supplying fly stocks and/or antibodies, and FlyBase for its wealth of information. We acknowledge the support of the Confocal Microscopy Unit at the Cajal Institute and at ISCIII. Finally, we thank Fundación Jose Maria Lopez Feliú and MICINN (Grants PI22CIII/00062 and PID2022-137751OA-I00) for funding.

## Additional information

### Funding

| Funder | Grant reference number | Author |
| --- | --- | --- |
| Ministerio de Ciencia e Innovación | PI22CIII/00062 | Sergio Casas-Tinto |
| Ministerio de Ciencia e Innovación | PID2022-137751OA-I00 | María Losada-Perez |

The funders had no role in study design, data collection and interpretation, or the decision to submit the work for publication.

### Author contributions

Sergio Casas-Tinto, Conceptualization, Funding acquisition, Writing – original draft, Project administration, Writing – review and editing; Nuria Garcia-Guillen, Data curation, Formal analysis, Methodology; María Losada-Perez, Conceptualization, Data curation, Formal analysis, Supervision, Funding acquisition, Validation, Investigation, Visualization, Methodology, Writing – original draft, Project administration, Writing – review and editing

### Author ORCIDs

Sergio Casas-Tinto ⬚ https://orcid.org/0000-0002-9589-9981
María Losada-Perez ⬚ https://orcid.org/0000-0002-6717-7762

Reviewer #2 (Public review): https://doi.org/10.7554/eLife.96890.4.sa1
Reviewer #3 (Public review): https://doi.org/10.7554/eLife.96890.4.sa2
Author response https://doi.org/10.7554/eLife.96890.4.sa3

## Additional files

### Supplementary files

MDAR checklist

### Data availability

All data generated or analysed during this study are included in the manuscript and supporting files.

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
