## [Editor Report · eLife Assessment]

In this work, the authors use a *Drosophila melanogaster* adult ventral nerve cord injury model extending and confirming previous observations. This **important** study reveals key aspects of adult neural plasticity. Taking advantage of several genetic reporter and fate tracing tools, the authors provide **solid** evidence for different forms of glial plasticity, that are increased upon injury. The significance of the generated cell types under homeostatic conditions and in response to injury remains to be further explored and open up new avenues of research.

---

## [Referee Report · Reviewer #2 (Public review)]

Summary:

Casas-Tinto et al., provide new insight into glial plasticity using a crush injury paradigm in the ventral nerve cord (VNC) of adult *Drosophila*. The authors find that both astrocyte-like glia (ALG) and ensheating glia (EG) divide under homeostatic conditions in the adult VNC and identify ALG as the glial population that specifically ramps up proliferation in response to injury, whereas the number of EGs decreases following the insult. Using lineage-tracing tools, the authors interestingly observe interconversion of glial subtypes, especially of EGs into ALGs, which occurs independent of injury and is dependent on the availability of the transcription factor Prospero in EGs, adding to the plasticity observed in the system. Finally, when tracing the progeny of glia, Casas-Tinto and colleagues detect cells of neuronal identity and provide evidence that such glia-derived neurogenesis is favored following ventral nerve cord injury, which puts forward a remarkable way in which glia can respond to neuronal damage.

Strengths:

This study highlights a new facet of adult nervous system plasticity at the level of the ventral nerve cord, supporting the view that proliferative capacity is maintained in the mature CNS and stimulated upon injury.

The injury paradigm is well chosen, as the organization of the neuromeres allows specific targeting of one segment, compared to the remaining intact and with the potential to later link observed plasticity to behavior such as locomotion.

Numerous experiments have been carried out in 7-day old flies, showing that the observed plasticity is not due to residual developmental remodeling or a still immature VNC.

Different techniques are used to observe proliferation in the VNC.

By elegantly combining different methods, the authors show glial divisions including with mitotic-dependent tracing and find that the number of generated glia is refined by apoptosis later on.

The work identifies prospero in glia as important coordinator of glial cell fate, from development to the adult context, which draws further attention to the upstream regulatory mechanisms.

Weaknesses:

The authors do not discuss their results on gliogenesis or neurogenesis in the adult VNC to previous findings made in the context of the injured adult brain.

The authors speculate about the role of glial inter-conversion for tissue homeostasis or regeneration, but no supportive evidence is cited or provided. Further experiments will be required to test the function of the described glial plasticity.

Elav+ cells originating from glia do not express markers for mature neurons at the analysed time-point. If they will eventually differentiate

or what type of structure is formed by them will have to be followed up in future studies.

Context/Discussion

Highlighting some differences in the reactiveness of glia in the VNC compared to the brain could reveal important differences in repair strategies in different areas of the CNS.

---

## [Referee Report · Reviewer #3 (Public review)]

In this manuscript, Casas-Tintó et al. explore the role of glial cell in the response to a neurodegenerative injury in the adult brain. They used *Drosophila melanogaster* as a model organism, and found that glial cells are able to generate new neurons through the mechanism of transdifferentiation in response to injury. This paper provides a new mechanism in regeneration, and gives an understanding to the role of glial cells in the process.

The authors have now addressed all my concerns.

---

## [Author Response]

The following is the authors’ response to the previous reviews.

**eLife Assessment**
In this work, the authors use a *Drosophila* adult ventral nerve cord injury model extending and confirming previous observations; this important study reveals key aspects of adult neural plasticity. Taking advantage of several genetic reporter and fate tracing tools, the authors provide solid evidence for different forms of glial plasticity, that are increased upon injury. The data on detected plasticity under physiologic conditions and especially the extent of cell divisions and cell fate changes upon injury would benefit from validation by additional markers. The experimental part would improve if strengthened and accompanied by a more comprehensive integration of results regarding glial reactivity in the adult CNS.

Thank you very much for your thoughtful comments and constructive feedback regarding our manuscript. We appreciate all the positive remarks on the significance of our findings on neural plasticity in this *Drosophila* adult ventral nerve cord injury model.

In response to your suggestion, we fully agree that the continuation of this project should address in detail cell fate changes with additional markers if available, or an ‘omic’ approach such as scRNAseq. Unfortunately, these further experiments are beyond the scope of this paper to describe the in vivo phenomena of cell reprogramming, and the cellular events that take glial cells to convert into neurons or neuronal precursors.

Additionally, we agree that the experimental part can be further improved by providing a more comprehensive integration of our results with current knowledge on glial reactivity in the adult CNS. We will revise the manuscript accordingly to include a deeper discussion of the broader implications of our findings and their alignment with existing literature.

Thank you again for your valuable input, which will undoubtedly enhance the quality of our work. We look forward to submitting the revised manuscript for your consideration.

**Public Reviews:**
**Reviewer #1 (Public review)**:Summary:Casas-Tinto et al. present convincing data that injury of the adult *Drosophila* CNS triggers transdifferentiation of glial cell and even the generation of neurons from glial cells. This observation opens up the possibility to get an handle on the molecular basis of neuronal and glial generation in the vertebrate CNS after traumatic injury caused by Stroke or Crush injury. The authors use an array of sophisticated tools to follow the development of glial cells at the injury site in very young and mature adults. The results in mature adults reveal a remarkable plasticity in the fly CNS and dispels the notion that repair after injury may be only possible in nerve cords which are still developing. The observation of so called VC cells which do not express the glial marker repo could point to the generation of neurons by former glial cells.Conclusion:The authors present an interesting story which is technically sound and could form the basis for an in depth analysis of the molecular mechanism driving repair after brain injury in *Drosophila* and vertebrates.Strengths:The evidence for transdifferentiation of glial cells is convincing. In addition, the injury to the adult CNS shows an inherent plasticity of the mature ventral nerve cord which is unexpected.Weaknesses:Traumatic brain injury in *Drosophila* has been previously reported to trigger mitosis of glial cells and generation of neural stem cells in the larval CNS and the adult brain hemispheres. Therefore this report adds to but does not significantly change our current understanding. The origin and identity of VC cells is still unclear. The authors show that VC cells are not GABA- or glutamergic. Yet, there are many other neurotransmitter or neuropetides. It would have been nice to see a staining with another general neuronal marker such as anti-Syt1 to confirm the neuronal identity of Syt1.

We thank the reviewer for the constructive comments and positive feedback. We concur that previous studies have demonstrated glial cell proliferation in response to CNS injury. In contrast, our study focuses on glial transdifferentiation that emerges as a novel phenomenon, particularly in response to injury. We found that neuropile glia lose their glial identity and express the pan-neuronal marker Elav. To investigate the identity of these newly observed elav-positive cells, we employed anti-ChAT, antiGABA and anti-GluRIIA antibodies to determine the functional identity of these cells, besides we stained them with other neuronal markers such Enabled, Gigas or Dac (not shown); however, our attempts yielded limited success. To address this, we have now included a discussion section exploring the potential identity of these cells, considering the possibility that they may represent immature neurons.

**Reviewer #2 (Public review):**
Summary:Casas-Tinto et al., provide new insight into glial plasticity using a crush injury paradigm in the ventral nerve cord (VNC) of adult *Drosophila*. The authors find that both astrocyte-like glia (ALG) and ensheating glia (EG) divide under homeostatic conditions in the adult VNC and identify ALG as the glial population that specifically ramps up proliferation in response to injury, whereas the number of EGs decreases following the insult. Using lineage-tracing tools, the authors interestingly observe interconversion of glial subtypes, especially of EGs into ALGs, which occurs independent of injury and is dependent on the availability of the transcription factor Prospero in EGs, adding to the plasticity observed in the system. Finally, when tracing the progeny of glia, Casas-Tinto and colleagues detect cells of neuronal identity and provide evidence that such gliaderived neurogenesis is specifically favoured following ventral nerve cord injury, which puts forward a remarkable way in which glia can respond to neuronal damage.Strengths:This study highlights a new facet of adult nervous system plasticity at the level of the ventral nerve cord, supporting the view that proliferative capacity is maintained in the mature CNS and stimulated upon injury.The injury paradigm is well chosen, as the organization of the neuromeres allows specific targeting of one segment, compared to the remaining intact and with the potential to later link observed plasticity to behaviour such as locomotion.Numerous experiments have been carried out in 7-day old flies, showing that the observed plasticity is not due to residual developmental remodelling or a still immature VNC.By elegantly combining different methods, the authors show glial divisions including with mitotic-dependent tracing and find that the number of generated glia is refined by apoptosis later on.The work identifies prospero in glia as an important coordinator of glial cell fate, from development to the adult context, which draws further attention to the upstream regulatory mechanisms.

We would like to thank the reviewer for his/her comments and the positive analysis of this work.

Weaknesses:The authors observe consistent inter-conversion of EG to ALG glial subtypes that is further stimulated upon injury. The authors conclude that these findings have important consequences for CNS regeneration and potentially for memory and learning. However, it remains somewhat unclear how glial transformation could contribute to regeneration and functional recovery.

This is an ongoing question in the laboratory and in the field. We know that glial cells contribute to the regenerative program in the nervous system, and molecular signalling in glial cells is determinant for the functional recovery (Losada-Perez et al 2021). Therefore, we include this concept in the discussion as the evidence indicates that glial cells participate in these programs. However, further investigation is required to clarify and determine the mechanisms underlying this glial contribution. To determine if glial to neuron transformation contributes to functional recovery, we would need to compare the recovery of animals with new VC to animals without VC, however, the molecular mechanism that produces this change of identity is still unknown, and therefore we are not able to generate injured flies with no new VC

The signal of the Fucci cell cycle reporter seems more complex to interpret based on the panels provided compared to the other methods employed by the authors to assess cell divisions.

We agree that Fly Fucci is a genetic reporter that might be more complex to interpret than EdU staining or other markers. However, glial cells proliferation is a milestone of this manuscript, and we used different available tools to confirm our results. We have revised this specific section to ensure that the text is clear and straightforward.

Elav+ cells originating from glia do not express markers for mature neurons at the analysed time-point. If they will eventually differentiate or what type of structure is formed by them will have to be followed up in future studies.

We fully agree with the reviewer, and we will analyze later days to study neuronal fate and contribution to VNC function.

Context/DiscussionThere is some lack of connecting or later comparing the observed forms of glial plasticity in the VNC with respect to plasticity described in the fly brain.Highlighting some differences in the reactiveness of glia in the VNC compared to the brain could point to relevant differences in repair capacity in different areas of the CNS.Based on the assays employed, the study points to a significant amount of glial ‘identity’ changes or interconversions under homeostatic conditions. The potential significance of this rather unexpected ‘baseline’ plasticity in adult tissues is not explicitly pointed out and could improve the understanding of the findings.Some speculations if ‘interconversion’ of glia is driven by the needs in the tissue could enrich the discussion.

We would like to thank the reviewer for these suggestions. We have changed the discussion to introduce these concepts.

**Reviewer #3 (Public review)**:In this manuscript, Casas-Tintó et al. explore the role of glial cell in the response to a neurodegenerative injury in the adult brain. They used *Drosophila melanogaster* as amodel organism, and found that glial cells are able to generate new neurons through the mechanism of transdifferentiation in response to injury. This paper provides a new mechanism in regeneration, and gives an understanding to the role of glial cells in the process.Comments on revisions:In the previous version of the manuscript, I had suggested several recommendations for the authors. Unfortunately, none of these were addressed in the author's revision.

We are sorry for this error. We apologize but we never received these comments. We have now found them, and we have incorporated these comments in the new version of the manuscript.

(1) Have you tried screening for other markers for the EdU+ Repo+ Pros- cells?

We have identified these cells as glial cells (Repo +), and not astrocyte-like glia (pros-). But we have not further characterized the identity of these cells. Our aim was to identify these proliferating glial cells as NPG (Neuropile glia), which are Astrocyte-Like Glia (ALG), as previous works suggest in larvae (Kato et al., 2020; Losada-Perez et al., 2016), or Ensheathing Glia (EG). To discard the ALG identity, we used prospero as the best marker. The results indicate that there are ALG among the proliferating population, but in addition, we also found pros- glial cells that were EdU positive. These cells are located in the interface between cortex and neuropile, where the neuropile glia position is described. The anti-pros staining indicated they were no ALG which suggest that they are EG.

There is no specific nuclear marker for EG cells, therefore we used FLY_FUCCI under the control of a EG specific promoter (R56F03-Gal4) to determine if the other dividing cells were EG. These results indicate that EG glia divide although their proliferation does not increase upon injury.

The R56F03 Gal4 construct is described as ensheathing glia specific by previous publications, including:

(1) Kremer M. C., Jung C., Batelli S., Rubin G. M. and Gaul U. (2017). The glia of the adult *Drosophila* nervous system. *Glia* 65, 606-638. 10.1002/glia.23115

(2) Qingzhong Ren, Takeshi Awasaki, Yu-Chun Wang, Yu-Fen Huang, Tzumin Lee. Lineage-guided Notch-dependent gliogenesis by *Drosophila* multi-potent progenitors. Development. 2018 Jun 11;145(11):dev160127. doi: 10.1242/dev.160127

To summarize, our results suggest that part of these proliferating glial cells are ALG and EG. Our results can not discard that a residual part of these proliferating cells are not AG nor EG.

(2) You mentioned that ALG are heterogenous in size and shape, does that mean that you may have different subpopulations of ALG? Would that also mean that only a portion of them responds to injury?

Yes, as in Astrocytes in vertebrates this population is highly heterogeneous. Currently there are no molecular tools to specifically identify these subpopulations and characterize their distinct roles. However, emerging research suggests that differences in size, shape, and potentially molecular markers could correlate with functional diversity. This implies that certain subpopulations of ALG may be more specialized or primed to respond to injury, while others may play roles in homeostasis or other processes. Understanding this heterogeneity will require advanced techniques such as single-cell RNA sequencing, spatial transcriptomics, or live imaging to unravel how these subpopulations contribute to injury responses and overall tissue dynamics.

(3) You mentioned that NP-like cells have similar nuclear shape and size to ALG and EG, while Ventral cortex cells have larger nuclei. Can you please show a quantification of the NP-like cells and Ventral cortex cells size, and show a direct comparison with ALG and EG cells to support those claims (images, quantification and analysis)?

We added a new supplementary figure with a graph showing nuclei size differences between VC and NP-like cells, and a diagram showing VC cell localization. Images in figure 2A-A’ and 2B-B’ show both types of cells with the same scale, additionally, NPG cells are shown in red (current expression of the specific Gal4 line). A direct comparison between EG and NP-like glia can be observed in Figure 3 as well.

Besides of size and localization, we conclude that VC and N-like cells present different molecular markers as VC are elav-positive and reponegative whereas NP-like cells are repo-positive elav-negative

(4) In Figure 2B, the repo expression is not very clear. I suggest using a different example to support the claim that NP cells are Repo+.

We have changed the color of anti-elav staining to facilitate visualisation

(5) Again, in Figure 2C, you need quantification and analysis to support the claim that you used nuclear shape and size to identify VC vs. NP like cells.

Quantification in point 3, criteria in Figure S1

(6) What is the identity of the newly formed neurons? Other than Elav, have you tried using other markers of neurons that are typically found in this area?

This question is of great interest and relevance. We have done great efforts to solve this open question and so far, our data suggest that these neurons might be in an immature state. In this last version of the manuscript, we included the results (Figure S1) with several different markers.

The molecular identity of these cell populations, glia and neurons, is currently under investigation.

Minor comments:(1) In the abstract, EG and ALG abbreviations are not introduced properly.

Thank you very much for noticing this missing information, we have now included it in the abstract.

(2) Please include a representation of the NPG somata location in Figure 1A.

We have included this information in the figure

(3) A schematic showing the differences between ALG and EG cells would be helpful as well.

We have included in the introduction references and reviews where other authors describe in detail the differences.

(4) In Figure 1 E, G, H- please indicated the genotype of the fly used in the panel as well as the cell type studied.

The complete genotype is included in the corresponding figure legend. We have added a simplified genotype in the figure for clarity.

(5) Please show the genotype used for images in Figure 2: ALG or EG specific drivers.

This information is included in the corresponding figure legend. We believe that it is better to keep the figure clean so we decided to keep the complete genotype, which is considerably long, only in the figure legend.